# Biology and Management of Pest *Diabrotica* Species in South America

**DOI:** 10.3390/insects11070421

**Published:** 2020-07-08

**Authors:** Guillermo Cabrera Walsh, Crébio J. Ávila, Nora Cabrera, Dori E. Nava, Alexandre de Sene Pinto, Donald C. Weber

**Affiliations:** 1ARS-SABCL/FuEDEI (Foundation for the Study of Invasive Species), Hurlingham B1686EFA, Argentina; 2EMBRAPA Agropecuaria Oeste, Dourados, Mato Grosso de Sul Caixa-postal 449, Brazil; crebio.avila@embrapa.br; 3Facultad de Ciencias Naturales y Museo, Universidad Nacional de La Plata, La Plata B1900FWA, Argentina; ncabrera@fcnym.unlp.edu.ar; 4EMBRAPA Clima Temperado, Pelotas, Rio Grande do Sul Caixa-Postal 403, Brazil; dori.edson-nava@embrapa.br; 5Centro Universitario Moura Lacerda, Ribeirão Preto, São Paulo 14076-510, Brazil; aspinn@uol.com.br; 6USDA-ARS Invasive Insect Biocontrol & Behavior Laboratory, Baltimore Avenue, Beltsville, MD 10300, USA; don.weber@usda.gov

**Keywords:** *Diabrotica speciosa*, *Diabrotica balteata*, *Diabrotica viridula*, rootworm management, maize pests, cucurbitacins, semiochemicals

## Abstract

The genus *Diabrotica* has over 400 described species, the majority of them neotropical. However, only three species of neotropical *Diabrotica* are considered agricultural pests: *D. speciosa*, *D. balteata*, and *D. viridula*. *D. speciosa* and *D. balteata* are polyphagous both as adults and during the larval stage. *D. viridula* are stenophagous during the larval stage, feeding essentially on maize roots, and polyphagous as adults. The larvae of the three species are pests on maize, but *D. speciosa* larvae also feed on potatoes and peanuts, while *D. balteata* larvae feed on beans and peanuts. None of these species express a winter/dry season egg diapause, displaying instead several continuous, latitude-mediated generations per year. This hinders the use of crop rotation as a management tool, although early planting can help in the temperate regions of the distribution of *D. speciosa*. The parasitoids of adults, *Celatoria bosqi* and *Centistes gasseni*, do not exert much control on *Diabrotica* populations, or show potential for inundative biocontrol plans. Management options are limited to insecticide applications and Bt genetically modified (GM) maize. Other techniques that show promise are products using *Beauveria bassiana* and *Heterorhabditis bacteriophora*, semiochemical attractants for monitoring purposes or as toxic baits, and plant resistance.

## 1. General Biology of South American Pest *Diabrotica*

The genus *Diabrotica* has over 400 described species [1], the majority of them neotropical, but only 7 species, plus six subspecies, are considered agricultural pests in the Americas [2]. Of these, only three species are considered agricultural pests in South America: *D. speciosa* (Germar) with subspecies *speciosa* and *vigens*, *D. balteata* (LeConte), and *D. viridula* (F.) (Figure 1). The genus *Diabrotica* is divided into three species groups: *virgifera*, *fucata*, and *signifera* [3,4]. However, studies on South American *virgifera* group species suggest that these groups are not as well defined as previously thought [5,6]. *D. speciosa* and *D. balteata* are in the *fucata* group, which is the group with the largest number of species. The species in this group that have been studied are polyphagous both as adults and during the larval stage. Another characteristic of the North American pest *Diabrotica* of the *fucata* species group is that they overwinter as adults and lack resistant stages to deal with harsh climatic conditions [2]. *D. viridula* is in the *virgifera* group, the same clade of the Northern, Western, and Mexican corn rootworms (*Diabrotica barberi*, *Diabrotica virgifera virgifera*, and *Diabrotica virgifera zeae*, respectively). The larvae of the North American species in the *virgifera* group feed exclusively on Poaceae [7], although the host range has been observed or tested for only a few of the species in the group [8]. The North American pest species in the *virgifera* species group are univoltine, or sometimes semivoltine, and possess diapausing eggs that allow them to overwinter in temperate climates or survive dry seasons in the subtropics [9,10], both situations during which the adult cannot find sustenance or survive the extreme conditions.

*D. speciosa* is distributed throughout South America, from agricultural patches in the temperate Patagonian steppes to the tropics, with the exception of Chile, and up to altitudes of over 2500 m above sea level [2,11] (Figure 2). It is the best studied *Diabrotica* species in South America due to its impact on many crops. The adult has over 132 recorded host species, in 24 different plant families [11, and literature therein]. Larval hosts are not as well known, but *D. speciosa* has at least five confirmed larval hosts: maize (*Zea mays* L.), wheat (*Triticum* spp.), Johnsongrass (*Sorghum halepense* Persoon), peanut (*Arachis hypogaea* L.), and potato (*Solanum tuberosum* L.). Another four plant species hosted full development in the laboratory [11,12,13,14,15]. However, the fact that larvae can develop on plant species in four families of three different orders suggests that there could be many more larval hosts that simply have not been discovered because of the hypogeous habit of the larva.

*D. speciosa* is documented in most crops in South America, but is considered mainly a horticultural pest as an adult, and a pest of potato, maize, and peanuts as larva [11,13,16]. Yet these generalizations are not without exceptions. In Brazil, this species is considered a pest of maize as a larva, and a minor pest as an adult as well [17,18]. It is also regarded as an important pest of potato during both the adult and larval stages, although this depends heavily on the cultivar [19]. In addition, the adult is also regarded as an important pest of seedlings and young plants of some extensive crops, such as soybeans, beans (*Phaseolus vulgaris*), cotton, sunflower, maize, tobacco, wheat, and canola [20,21,22], and, curiously, of table grapes [23] (Table 1).

*D. balteata* is found from subtropical North America through Central America and Caribbean islands including Cuba, Hispaniola, and Puerto Rico, to South America, although its distribution in South America is limited to Venezuela and Colombia [2,24], where it can occur at altitudes ranging from 0 to 2000 m [25]. However, there is insufficient data to infer species distribution patterns in either country. The adult of *D. balteata* also has an extremely wide range of host plants, as it has been documented on over 140 plant species [26]. There is a more conservative estimate of 50 species in 23 families, with a preference for plants in the Cucurbitaceae, Rosaceae, Fabaceae, and Brassicaceae [27]. The *D. balteata* adult is considered a pest on squash (*Cucurbita* spp., Cucurbitaceae), several bean species (*P. vulgaris*, *Glycine max*, *Mucuna pruriens*, and *Vigna unguiculata*, Fabaceae), lettuce (*Lactuca sativa*, Asteraceae), sugar cane (*Saccharum officinarum*, Poaceae), and potato [28]. Adults are also implicated in the transmission of the tomato brown rugose fruit virus (Tobamovirus, ToBRFV) to *P. vulgaris* [29], and other viruses of *P. vulgaris* and calapo (*Calopogonium mucunoides* Desv.) [30,31]. Larval damage is reported only from Colombia, where this species is known to attack beans, but as considered a minor problem [32], maize, on which it can be locally problematic [33,34], and peanuts, on which it is considered among the 10–12 worst pests in Colombia [35] (Table 1). The larva has also been reported to attack sweet potato in the USA [36], although not in South America. Yet, the fact that these hosts are also from three families in three orders suggests that there could be many more larval hosts as well. In addition, phylogenetic studies indicate *D. speciosa* and *D. balteata* are sister clades [37].

*D. viridula* is distributed from Mexico to northern Argentina, and apparently absent in Uruguay and Chile, except on Easter Island, where it was introduced [2,13,38] (Figure 3). Like *D. balteata*, its distribution is primarily tropical and subtropical. The *D. viridula* adult is considered a minor pest of beans in Peru [39], while the larva is considered locally important on maize in Central America and Peru [40]. In greenhouse tests, both the larvae and the adults of this species were able to transmit maize chlorotic mottle virus (MCMV) to maize, and they are assumed to be one of its vectors in the field [41]. *D. viridula* is also assumed to be an important, albeit new, pest of maize roots in Argentina, Paraguay, and Brazil [11,42], but its damage cannot be differentiated from that of *D. speciosa*. Studies to clarify what proportion of the damage is owed to each species (e.g., collections of larvae directly in the field) have not been done. The larva has been found feeding on maize roots only, and in the laboratory, it developed successfully on wheat as well, but not on any of the species tested from outside the Poaceae, suggesting it is stenophagous during the larval stage [2,43]. As an adult it is polyphagous, albeit reduced to fewer hosts than *D. speciosa* and *D. balteata*, as it has been recorded only on 21 plant species in the Poacae, Cucurbitaceae, and Asteraceae [11] (Table 1). Yet, similarities with the North American species in the *virgifera* group end here, as *D. viridula* eggs do not diapause. This species was reared in the laboratory for many generations, and the eggs never expressed any delay in hatching at optimal developmental temperature (8 ± 1 days at 25 ± 1 °C), regardless of previous photoperiod and temperature conditions (0 ± 1, 5 ± 1, 13 ± 1 °C; 10:14, 12:12, 14:10 h (L:D)) [13,43,44]. Eggs from field-collected adults, including overwintering adults, expressed no delay in hatching either [44].

Evidence indicates that the three South American pest *Diabrotica* overwinter as adults, are multivoltine, and do not have diapausing eggs. A reproductive diapause has been observed for *D. speciosa*, at least for the populations from the temperate and higher subtropical areas, but the fact that it could be overridden by manipulating temperature and light hours suggests it may not exist in the lower latitudes [44].

## 2. Control of South American *Diabrotica*

As the North American corn rootworms in the *virgifera* species group overwinter as diapausing eggs, are univoltine, and have a narrow larval host range limited to maize and a few grasses, their life cycle is tightly coupled to the phenology of one or very few annual host species. This provides opportunities for the use of different management strategies to reduce damage levels on susceptible crops, such as crop rotation and manipulation of sowing dates [45,46], expected density functions based on preceding density data [47], and anticipation of adult appearance through degree-day models [48]. Also, as the eggs are found anywhere in the soil from before the crop is planted, different tillage techniques could be applied to hinder the larvae from reaching the roots, for instance, compacting the soil between rows, thus affecting neonate larval movement [49]. Furthermore, factors behind the recommencement and completion of embryonic development after winter in univoltine *Diabrotica* are fairly well understood, so it is possible to estimate a “fixed point” (or interval) for the conclusion of embryonic development of the egg bank laid during the previous season in any given area [50]. However, none of these options have been developed for multivoltine species.

The field biology of the multivoltine species of the North American pest *Diabrotica* is also relatively well understood. Yet, in contrast to the univoltine species, predicting the incidence of the multivoltine species is not easily achieved. The only predictive tool of which the authors are aware has been used to calculate the probable damage of *Diabrotica undecimpunctata howardi* Barber on peanuts. This index used data such as soil texture, soil drainage class, planting date, cultivar resistance, and field history of rootworm damage to determine when to apply soil insecticides. Although the index recommended insecticide applications for 98.5% of the fields that actually needed insecticide treatment, it also recommended treatment for over 50% of fields that did not need it [51].

Although it is certain that the South American pest *Diabrotica* are multivoltine, seasonal reproductive patterns are not well known for these species. Soil and air temperatures were used in a linear degree-day model in laboratory and greenhouse experiments, to predict the occurrence of adults of *D. speciosa* [52]. The authors found that soil and air temperatures provided a significantly different prediction of insect occurrence than those observed experimentally. However, the prediction of occurrence based on soil temperature was more accurate than when the air temperature was used. One study in Argentina based on teneral collections in different regions suggests that the single most important determinant for the emergence of *D. speciosa* adults was weekly average temperatures above 13 °C. Due to this, in the temperate distribution areas of *D. speciosa*, there could be around three generations a year, and in subtropical regions, no fewer than five. However, no obvious or discrete voltinism pattern could be observed, expressing, to all practical effects, continuous generations [53]. What is known of the reproductive biology of the other two pest species suggests the same may be expected for them. Under the circumstances, it may be feasible to predict the appearance of a first generation after winter, in the areas where larval development might be temperature-limited, but such prediction may not be accurate enough to calculate planting dates, and certainly not apt to determine predictable cohorts. The practical implications of this study were that the life history pattern of this pest seems to leave few management alternatives. In the temperate regions of this species’ distribution, early planting of maize could ensure that the first generations of larvae encounter more mature, and thus less susceptible stages of the crop. Other than this, the seasonal dispersion and unpredictability of *D. speciosa* outbreaks suggest that the only pre-emptive action available to protect maize crops from this pest is to plant Bt maize [53].

As mentioned above, the damage on maize from *D. speciosa* larval feeding cannot be differentiated from that of *D. viridula*, so control measures implemented for the control of *D. speciosa* larvae apply to *D. viridula* as well (Figure 4). In addition, the vast majority of references to research on *Diabrotica* spp. control in South America apply to *D. speciosa*, or are general for several agricultural pests.

### 2.1. Chemical Control

Most control efforts in agriculture in South America are aimed at foliar pests and stem borers. There are published recommendations for treatment thresholds based on adult *Diabrotica* sampling protocols and foliar damage rates for beans and soybeans, respectively [54,55]. Yet, some control measures for root-feeders have been attempted, mainly seed treatments, in-furrow spraying, and granular pesticide applications [56,57]. There are no published calculations of the input of pesticides used for maize, beans, and potato, but they are generally considered to be high [58]. In Brazil there are 129 pesticides registered for *D. speciosa* in maize, potatoes, and beans, including foliar sprays, in-furrow, seed treatments, and four biological products based on *Beauveria bassiana* and one based on *Heterorhabditis bacteriophora* [58] (Table 1).

References for chemical control of *Diabrotica* in Argentina, Peru, and Uruguay follow more or less the same tendency of recommending several broad spectrum pesticides for adult control: chlorpyrifos, methomyl, other carbamates, fenitrothion, and several pyrethroids [59,60]. We have not found references to chemical control of larvae, and in fact concern for larval damage from *Diabrotica* is relatively recent, and all root-damaging insects are combined insofar as treatment actions are concerned. Their control has been trusted essentially to seed treatments with carbamates, neonicotinoids such as clothianidin, thiamethoxam, and imidacloprid, recently combined with diamides (cyantraniliprole and chlorantraniliprole), and genetically modified (GM) maize [61,62] (Table 1). However, seed treatments have been reported to be inefficient ways of controlling *D. speciosa* larvae on maize in Brazil [63]. Several authors reported that the most effective treatments are liquid in-furrow applications with organophosphates and phenylpyrazole insecticides in maize [64,65], and neonicotinoids for potatoes [66]. Granular applications also showed promise, but are not recommended due to technical limitations related to the cost and efficiency of granular applicators, and toxicity risks [67]. Finally, silicon applications have been reported to help decrease adult damage from *D. speciosa* and *Liriomyza* spp. (Diptera: Agromyzidae), leaf miners in organic potatoes [68].

Insecticides that interfere with the development of immature forms of insects (insect growth regulators (IGR)) can also cause a sterilizing effect on adult Coleoptera, affecting their fecundity and egg viability [69,70]. *D. speciosa* adults fed bean leaves treated with the IGR lufenuron showed a significant reduction in fertility and egg viability [71,72]. This deleterious effect on the progeny might reduce their biotic potential in the field, without using soil treatments (Table 1), although this has yet to be confirmed.

References to the evolution of insecticide resistance in South American *Diabrotica* are absent in the literature. However, this does not mean that it does not occur, but perhaps that it has not been studied.

### 2.2. Genetically Modified Crops

GM crops are one of the most widespread options for insect management in South America. GM maize, cotton, and soya are widely planted in Brazil and Argentina, the second and third countries with the largest productions of GM crops in the world, respectively, after the USA [73]. GM maize containing the Cry3Bb1 gene has been available in both countries since 2010 [57]. Up to 90% of the maize sown in Brazil is GM [57], and 96% in Argentina [73], mostly for control of Lepidoptera. Field tests showed that root damage levels were, without exception, lower than economic threshold, while yield was 2 to 5% higher than that of susceptible maize of the same variety [57]. Several lines of maize containing the Cry3Bb1 and the Cry1Ab genes were tested in greenhouse feeding tests with *D. speciosa* in Argentina in 2004. A 15-stage rating system was applied, which revealed that both events afforded some protection from larval damage compared to that seen in their conventional near-isolines. In the tests, however, the lines with the Cry3Bb1 gene suffered significantly lower damage levels (Cabrera Walsh, unpublished). Other countries in South America show a similar pattern, such as Paraguay (virtually 100% of its maize, [74]), and Uruguay, where there are no official data, but the area cultivated with GM maize is estimated at 86% [75]. This situation is not observed in Colombia, with only 31% of its maize crop being GM [76], Peru, where there is a moratorium on GM crops until 2021 [77], or Bolivia, where GM maize has recently been approved for planting, but its level of adoption remains unreported [78] (Table 1).

A new Bt protein, aimed especially for the control of *D. speciosa* larvae, was made available to maize growers during the 2013–2014 season, especially in south-central Brazil. This transgenic cultivar contained two Bt proteins expressed in the aerial parts aimed at caterpillars, and another specific protein (Cry3Bb1) for the control of *D. speciosa* larvae. Silva et al. [79] evaluated the efficiency of the Cry3Bb1 protein present in maize for the control of *D. speciosa* larvae, confirming higher productivity than that of the susceptible maize, and fewer larvae in the rhizosphere. Gallo [80] also evaluated the efficacy of corn genotypes that express the Cry3Bb1 protein for the control of *D. speciosa* larvae, and reported that both genotypes tested were effective in reducing corn root damage compared to that of other genotypes free of this toxin.

Potatoes expressing both the Cry3A and Cry1Ia1 genes were developed, field tested, and deemed to be effective to control *D. speciosa* [81]. However, these potato varieties were never commercialized.

### 2.3. Plant Resistance

Damage of *D. speciosa* on potatoes can be locally severe, both from adult damage to the aerial parts, and larval damage to the roots and tubers [82]. Work has been done to promote natural resistance in potato. This can come from chemical defenses, such as leptins (which are insecticidal peptides) and natural glycoalkaloids, which can confer resistance to both adults and larvae. Furthermore, the density and type of trichomes expressed by the plant can influence adult feeding behavior. These defense mechanisms can be selected from different cultivars, or incorporated from different species of wild potatoes [82,83,84].

In South Carolina (USA), sweet potatoes have been evaluated for *D. balteata* resistance [85]. In Florida (USA), where *D. balteata* is a key pest of lettuce, resistance has been evaluated based on the effective expression of latex upon injury [86,87]. Beans can also be selected for trichome expression to confer defoliation resistance to many pests, not only *Diabrotica* spp. [88,89].

Native resistance in maize to South American *Diabrotica* has not been tested, but it should be explored given the high number of native maize varieties in South America. Experiments in the US indicate that some maize genotypes expressed native antibiosis that reduced *D. virgifera virgifera* feeding significantly, as compared to that in the more susceptible genotypes. Damage was still higher than for a control GM maize, but larval development was not significantly different between the GM control and the more resistant maize genotypes [90] (Table 1).

Although not actually a form of plant resistance, intercropping shows some promise as a management option as well. There is some evidence of reduced incidence and damage from several bean pests, including *Diabrotica* sp., on *P. vulgaris*, based on intercropping with sugar cane in Colombia [91]. Intercropping beans with banana, maize, and other crops has shown mixed, although often favorable results in Central America [92,93] (Table 1).

### 2.4. Biological Control

In spite of the large number of species in the *Diabrotica* genus, and how widespread several of them are, only five species of parasitoids are known for the whole genus [94,95]. This is not the result of a lack of survey efforts, since many entomologists have surveyed for parasitoids and pathogens for many years throughout the Americas, and only one new species was detected in 60 years ([94], and literature therein). The scarcity of parasitoids of adults in the genus has been hypothesized to be due to the accumulation of cucurbitacins in fatty tissues [96,97]. These triterpenes are frequent in the Cucurbitaceae, common feeding hosts of adults in the genus, and are known to have antifeedant properties, but act as feeding stimulants for *Diabrotica* spp. [98,99]. There are no references of predators or parasitoids of larvae of South American species of *Diabrotica* [94]. However, based on the wide range of predators detected for *D. virgifera virgifera* in North America [100,101], it is to be expected that there are egg and larval predators of South American *Diabrotica* as well, which are yet to be discovered. *Diabrotica virgifera virgifera* larvae were found to have potent hemolymph defenses against predators [102,103], which may also be present in other *Diabrotica* spp.

Two adult parasitoid species, *Centistes gasseni* (Hymenoptera: Braconidae) and *Celotoria bosqi* (Diptera: Tachinidae), are known to parasitize *D. speciosa* and *D. viridula*, but with extremely low incidences in the latter. *Celatoria compressa* (Diptera: Tachinidae) is known to parasitize *D. balteata* in North and Central America, with no records for South America [104,105,106,107]. Other than these, at least 10 generalist predators have been recorded for adult *D. speciosa* [108].

Natural parasitism levels in *D. speciosa* have been reported between 1 and 28%, and on rare occasions over 30% [105,106]. Furthermore, the higher levels of parasitoidism are always recorded toward the end of the growing season, when most of the crop damage is done, suggesting that natural control levels are of minor importance to pest management [108]. It seems unlikely that biological control with macro-organisms will provide any significant relief to agriculture, or to have much potential at this stage for inundative biocontrol plans, given their low reproductive rate, comparatively long development, and dependence on laboratory-reared adults. However, new advances in parasitoid rearing could change this situation in the future [109].

Biological control with pathogens and nematodes offers a different outlook, with several promising laboratory and greenhouse results. Several strains of *Beauveria bassiana*, *B. brongniartii* (Hypocreales: Cordycipitaceae), and *Metarhizium anisopliae* (Hypocreales: Clavicipitaceae) were effective in controlling *Diabrotica virgifera virgifera* larvae for up to 21 days after application [110]. Similar results have been obtained for South American species. In Brazil, the microbial control of *D. speciosa* larvae with entomopathogenic fungi or nematodes is considered to have great potential because the soil is a relatively stable environment in terms of temperature and humidity, especially in no-till farming [111]. Argentine strains of *M. anisopliae* and *B. bassiana* killed third instars of *D. speciosa* in the laboratory [112]. Brazilian strains of *Isaria fumosorosea* (Hypocreales: Clavicipitaceae) and *Purpureocillium lilacinum* (Hypocreales: Ophiocordycipitaceae) killed eggs of these species, also in the laboratory [113]. Twenty strains of entomopathogic fungi (*B. bassiana*, *M. anisopliae*, and *P. lilacinum*) were colonized as endophytes in tobacco from northern Argentina. However, feeding tests on *D. speciosa* adults with the treated plants showed no significant differences with endophyte-free plants [114] (Table 1).

A few studies have also been translated to field conditions for biological control of *D. speciosa* in production systems [113,115]. Promising results were obtained with the strain of *B. bassiana* ESALQ PL63, used in seed treatments, which decreased the defoliation caused by *D. speciosa* adults in beans for more than three weeks after seeding [116]. Similar results were obtained in maize when the soil was treated with *Pseudomonas* (Pseudomonadales: Pseudomonadaceae) [117] and *Bacillus pumilus* [118].

Rhabditid nematodes (Steinernematidae and Heterorhabditidae) have been studied to control corn rootworms for decades, often with promising results. In the field, *Heterorhabditis bacteriophora* Poinar (Rhabditida: Heterorhabditidae) was as effective as tefluthrin in controlling *Diabrotica virgifera virgifera* in corn crop [119] and with a long residual action in the soil [120,121]. Seventeen native and exotic entomopathogenic nematode isolates (Steinernematidae and Heterorhabditidae) were tested against *D. speciosa* under laboratory and greenhouse conditions in Brazil on eggs, third (last) instars, and pupae. High mortality rates were obtained with *Heterorhabditis* sp. RSC01 and JPM04, *Steinernema glaseri*, and *Heterorhabditis amazonensis* on larvae and pupae, while eggs were unaffected [121]. These nematodes are considered to have great potential to control *D. speciosa* in irrigated maize and potatoes [122] (Table 1).

Maize roots attract entomopathogenic nematodes with (E)-β-caryophyllene when fed upon by *D. balteata* and other *Diabrotica*, and production of this chemical is enhanced by certain root-colonizing bacteria [123]. Furthermore, Jaffuel et al. [120] have shown that *Heterorhabditis bacteriophora*, encapsulated in durable alginate-based beads, effectively controlled *D. balteata* larvae in greenhouse tests.

Mermithidae have been cited quite often from *D. speciosa* adults [95,124,125] as well as *D. balteata* [126], but they are generally considered to be too difficult to mass rear, so are probably not feasible biocontrol agents [122].

### 2.5. Semiochemicals

*D. speciosa* females exhibited calling behavior similar to that described for *Diabrotica virgifera virgifera* [127,128]. Nardi [129] studied the sexual behavior of *D. speciosa*, observing mating from the third day after the emergence of the females. Mating was concentrated from 6 p.m. to midnight. Based on these studies, it became evident that the sexual behavior of *D. speciosa* was well defined, and that sexual attraction was probably mediated by a sexual pheromone produced by females. Yellow plastic cups coated with an adhesive and baited with females, especially virgin females, attracted males. Males of different age or reproductive state enclosed in the same cups did not attract females nor males [130]. Y-tube olfactometer and GC-EAG tests showed that males of *D. speciosa* were attracted by volatile compounds emitted by females. However, this compound has not been identified yet. Male volatiles were not attractive to either sex [128].

The female-produced sex pheromone for *D. balteata* is (*R*,*R*) 6,12-dimethylpentadecan-2-one [131,132]. Although stereospecific syntheses have been published [133,134,135], the racemic mixture is attractive, based on the single active stereoisomer [132]. It was attractive to males in the field in South Carolina (USA) and potentially useful for monitoring and management [136], and is commercially available in the USA [137].

The floral compound 1,2-dimethoxybenzene, one of the main floral volatiles of *Cucurbita maxima*, was found attractive to *D. speciosa* adults. Traps baited with TIC (1,2,4-trimethoxybenzene + indole + trans-cinnamaldehyde) and VIP (veratrole + indole + phenylacetaldehyde) also attracted *D. speciosa* adults, but less effectively [138]. Although 1,2-dimethoxybenzene is a very abundant and well-known floral component, it had not been reported as an attractant for *Diabrotica* spp. before, suggesting *D. speciosa* has a unique response pattern for floral volatiles [130]. Ensuing studies showed that the attractiveness of this compound was quite specific, as none of the analogs tested were attractive to adults [139].

Olfactometer tests with seedlings have shown that CO_2_ and unidentified host specific root compounds from maize and oat seedlings were attractive to *D. speciosa* larvae. Wheat, beans, and soybean seedlings also elicited a response, albeit less vigorous [140]. Johnson and Gregory [141] reported that CO_2_ is involved in general orientation, while specific compounds are involved in fine orientation toward the host plant roots. In any case, Nardi [129] argued that *D. speciosa* larvae have a very limited capacity for movement and host location, and it is the gravid female that chooses the host plants, suggesting there may not be much of a future for *D. speciosa* management in larval attractants. Regardless, this information could be useful in future research on chemical communication and development of management techniques for this species [142], but to date, no pheromones or floral attractants have been synthesized for practical uses.

As mentioned above, there are many references to the attractant and/or arrestant effects of cucurbitacins to adult *Diabrotica* spp. Several pest management tactics have been implemented based on the phagostimulatory effect of cucurbitacins on diabroticine beetles. These include lacing bitter cucurbit roots or fruit with an insecticide [128,143,144], using the roots or fruits in traps for monitoring and collecting Luperini [10,145,146,147,148], bitter cucurbit juice formulations combined with fungal pathogens [149], and in toxic baits [150,151,152,153,154,155]. Cucurbitacins have also been included as baits in traps for monitoring purposes [10,145,147,156,157].

Although it is clear that cucurbitacins are phagostimulants, there were contradictory reports as to them being volatile kairomones as well (see [147] for a full discussion on the subject). The difference is that volatile kairomones have the power to attract the recipient from a distance, whereas arrestants cause the recipient to remain only after the individual has made contact with the compound. These characteristics potentially provide different applications, because whereas an arrestant in a toxic bait can drive the target insect to ingest the insecticide, it will not attract it from a distance, precluding its use in traps. Kairomones, on the other hand can serve both purposes if they are phagostimulants as well, as is the case with cucurbitacins. Field experiments in Argentina showed that only males of *D. speciosa* were attracted from a distance to cucurbitacins (*ca.* 20 m), whereas for females these compounds acted only as arrestants, and to a lesser degree than for males [11,148]. This indicates that control or monitoring devices reliant on distance attraction to bitter cucurbit extracts would function exclusively on *D. speciosa* males. However, the wide dispersal of a toxic bait based on cucurbitacins promoted encounters and control of both sexes within the treated area, without any significant non-target effects [155,158] (Table 1).

## 3. Conclusions

*Diabrotica* management in South America has been stagnated for several years. Apart from insecticide applications, the major innovation of applicable use of the last 30 years has been the introduction of GM maize. However, other techniques that show promise must continue to be explored, such as the use of toxic baits with semiochemical attractants to suppress adult populations and for monitoring purposes, IGR insecticides aimed at adults to reduce their progeny, development of plant resistance, and biological control using *Heterorhabditis* nematodes and entomopathogenic fungus against larvae. Insecticide + cucurbitacin baits also deserve a special mention, because this combination has proved to be an effective technique that probably warrants further development.

Pest *Diabrotica* in South America are widely regarded as important, but usually are not differentiated from other foliar pests or root-feeders when it comes to management. Farmers do not identify them among the worst pests, and seldom deploy specific control measures for these beetles, except for potatoes in Brazil, where producers consider *D. speciosa* to be the main pest. Yet, the actual impact of the larvae of *D. speciosa* and *D. viridula*, especially on maize, may not be properly assessed, and until that is done, we cannot be sure of the real importance of these pests.

## Figures and Tables

**Figure 1 insects-11-00421-f001:**
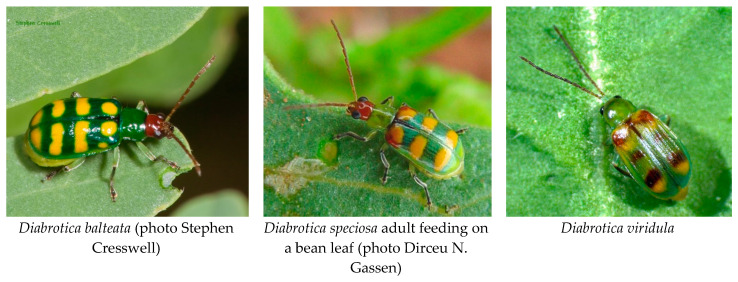
Photographs of the adult of the three species of pest *Diabrotica* from South America.

**Figure 2 insects-11-00421-f002:**
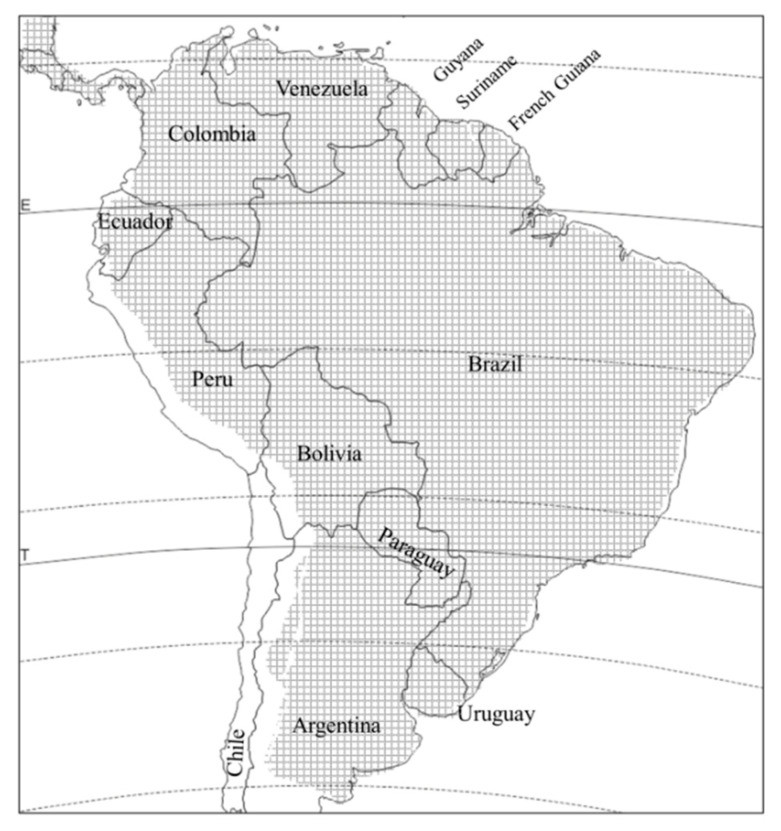
Distribution of *Diabrotica speciosa* in South America (crosshatched area).

**Figure 3 insects-11-00421-f003:**
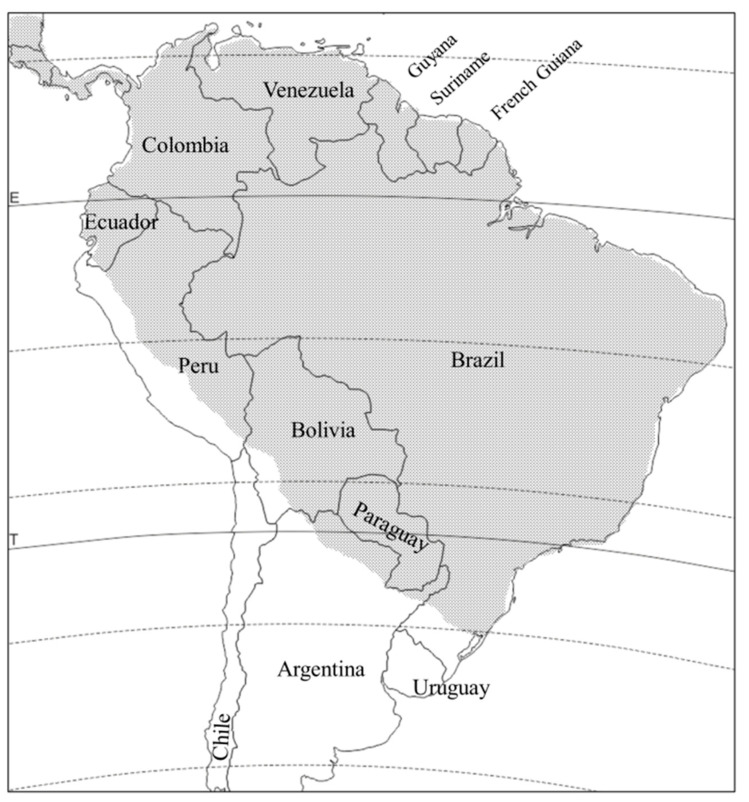
Distribution of *Diabrotica viridula* in South America (stippled area).

**Figure 4 insects-11-00421-f004:**
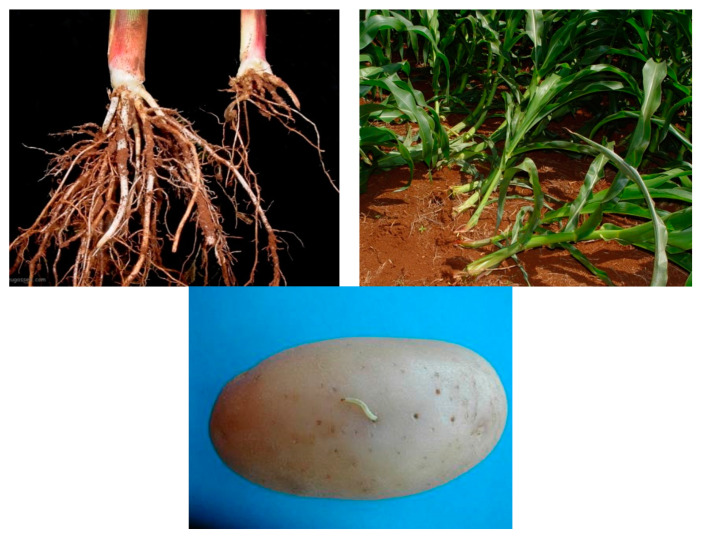
Top, typical damage on maize roots and lodging caused by *D. speciosa* and *D. viridula* larvae. (photos by Dirceu N. Gassen); below, *D. speciosa* larva on potato with typical pinprick damage (photo by Pablo Lanzetta).

**Table 1 insects-11-00421-t001:** Main crops attacked by the South American pest *Diabrotica* species, and current and potential control methods.

	*D. balteata*		*D. speciosa*		*D. viridula*		Control Methods		Promising Control Methods	
Host Crop	Adults	Larvae	Adults	Larvae	Adults	Larvae	Adults	Larvae	Adults	Larvae
beans	x	x	x		x		Cb, Op, Nn, Py ^1^		intercropping plant resistance	
cucurbits	x		x				Cb, Op, Nn, Py		cucurbitacin baits	
maize		x	x	x		x	Cb, Op, Nn, Py	Bt maize seed treatment (Nn, Cb, Di) ^1^	silicon cucurbitacin baits	IGR ^1^ seed treatment with fungi, plant resistance, nematodes
peanuts		x	x	x			Cb, Op, Nn, Py			
potatoes	x		x	x			Nn		plant resistance	plant resistance, nematodes
soybeans			x				Cb, Op, Nn, Py			
tobacco			x				Cb, Op, Nn, Py			

^1^ Cb, carbamates; Op, organophosphates; Nn, neonicotinoids; Py, phenylpyrazole; Di, diamides; IGR, insect growth regulators

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
