# Peer review of "Biology and Management of Pest Diabrotica Species in South America"

_insects, 2020, doi:10.3390/insects11070421_

Round 1

Reviewer 1 Report

Manuscript Walsh et al. ‘Biology and management of pest Diabrotica species in South America’ is a useful review about the major Diabrotica pests in South America. There are a few points that the authors can probably include in the text, such as cases of resistance evolution to insecticides, if any, and if none, what are the reasons? No evaluation done? Or is it because D. speciosa is polyphagous pest that moves from crop to crop? Also, has the use of Bt corn targeting Lepidoptera made Diabrotica become a problem due to the reduced use of insecticides?

Below are some comments to improve the ms.

Lines 17, 35: replace ‘recognized’ by ‘identified’

Line 21-22, maybe it sounds better if rewritten: ‘The larvae of the three species are major maize pests, but D. speciosa larvae also feed on potatoes and peanuts, while D. balteata larvae feed on beans and peanuts.’

Line 49: add reference Clark and Hibbard (2004, Env. Entomol., https://doi.org/10.1603/0046-225X-33.3.681)  

Line 55: please define m.a.s.l.

Line 58: please mention the names of the plants

Line 60, 86: add ‘it’ before ‘suggests’

line 64: remove ‘a’ before larva

Line 55, 74: 2000 or 2 thousand? Please keep it consistent throughout the manuscript

Line 75: ‘conservative estimate’

Line 91: PRIMARILY instead of ‘eminently’

Lines 96-97: ‘…and studies to clarify this statement ...’

Line 103: ‘…as D. viridula eggs do not diapause.’

Line 105: replace ‘independently’ by ‘regardless’; What developmental temperatures? How many days to hatch? Please specify

Line 121: ‘reaching’ instead of ‘accessing’

Line 122: …thus affecting neonate larvae movement.

Line 123: delete ‘blocking their access to the maize roots’

Line 152: …of this study WERE…

Line 163: It’d be useful to add a paragraph about any case of D. speciosa resistance to insecticide, and if not, why there is no cases, no research done? Polyphagous pest that moves from crop to crop, thus ‘diluting’ any resistance outbreak? Any reference?

Line 177: Are these ‘recent’ cases of Diabrotica spp. damage (on maize??) the result of Bt corn used to control Lepidoptera (since 2007-2008 in Brazil?); thus, Diabrotica would be a secondary pest that became primary pest after control of, for instance, Spodoptera frugiperda, due to the reduced use of insecticides? Please add a paragraph on this topic, if possible.

Line 177: replace enveloped by ‘INCLUDED’

Line 187: leaf MINERS

Lines 193-194: …without USING SOIL INSECTICIDES.

Line 208: data instead of ‘figures’

Lines 211-212: …but no data are available.

Line 220: …and reported that both genotypes…

Line 227: …and larval damage to roots and tubers.

Line 228: …this can BE from…

Line 238: …should be EXPLORED GIVEN THE HIGH NUMBER OF NATIVE MAIZE VARIETIES IN SOUTH AMERICA.

Line 245: awkward sentence, consider revising run on sentence; …sugar can in Colombia. Additionally, low Diabrotica damage on beans has been reported when intercropping with banana, maize,…

Lines 261-265, rewrite sentence: Two adult parasitoid species, Centistes gasseni (Hymenoptera: Braconidae) and Celotoria bosqi (Diptera: Tachinidae), are known to prey on D. speciose and D. viridula, but with extremely low incidence on the latter. Celatoria compressa (Diptera: Tachinidae) is known to prey on D. balteata in North and Central America, with no records for South America. Furthermore, at least 10 generalits…

line 267: …higher levels OF PREDATION are always recorded…

Line 268: ..most of the PLANT damage?

Line 273: in THE future.

Line 319: …Males of DIFFERENT age or reproductive STAGE…

Line 330: …was found ATTRACTIVE to D. speciosa..

Line 334: replace ensuing by ‘additional’

Line 336: …attractive to adults.

Line 343: delete ‘for them’; rewrite ‘…suggesting that more research on larval attractant needs to be performed.’

Lines 344-346: this sentence might not be necessary

Author Response

First off, I would like to thank the referees on behalf of the other authors and I, for the thorough and helpful revision. We have introduced almost every modification suggested by them, which you will find in track changes throughout the manuscript. We also added a reference suggested by ref. No. 1, so we had to move all the literature numbering one place up. These we did not mark so as not the crowd the manuscript out.

We did leave some of the suggested corrections out, or complied partially. They follow:

Ref. No. 1

Line 97-98, the studies we observed lacking are not about the statement, of which we have no doubt, but about what proportion of the damage on maize is caused by each Diabrotica species. We have rewritten the line a bit to make it clearer.

The referee asked for references on cases of D. speciosa insecticide resistance evolution, and if there weren’t, why. It’s a brilliant point, and we did, indeed, search for references on the matter, and found none. There is plenty of work on IRM, both for insecticides and GM maize, but not because it has been reported already, but in anticipation of its appearance. We only added a brief sentence about it (line 197).

The reviewer also asks if the pest Diabrotica may have become more damaging on maize due to a reduction in the use of pesticides after the introduction of Bt corn for lepidopteran control (Line 178). That is not what we meant, we meant literally what the sentence says: concern for Diabrotica damage is recent, not awareness of their damage, only concern. In fact, there is no reason to assume that it has increased. Until quite recently, farmers and extensionists enveloped all root feeders in one guilty party that received the same treatments (line 179), and everybody knew Diabrotica was among them, and was probably important, but didn’t really care to gauge the damage of rootworms, wireworms and scarab grubs separately. The story goes like this: 15 years ago the first GM maize designed for larval Diabrotica control arrived in South America, and the large agricultural corporations were eager to peddle them in Argentina and Brazil, so they sponsored quite a lot of research on South American pest Diabrotica (most authors of this manuscript were involved in this research in one way or another), effectively creating a Diabrotica craze in the region. I know this story well, I was part of it, but I can’t see how we could really put in the manuscript: “until M…anto, S…enta and P…eer decided to sell their anti-rootworm seed to South American farmers, nobody paid particular attention to Diabrotica larvae”, Ha!

In line 348, the reviewer indicates we should change “suggesting there may not be much future for D. speciosa management in larval attractants.”, for “suggesting that more research on larval attractant needs to be performed.”. We have decided against it because it changes the meaning of the sentence completely. What Dr. Cristiane Nardi suggested was that larval attractants might not be a useful, or applicable, research road in the case of Diabrotica speciosa. She might be wrong -although I rather agree with her-, but that doesn’t change the fact that that’s what she suggested.

I think that’s all, the remaining corrections have been done.

Reviewer 2 Report

Walsh et al present a deep review about the biology and management of Diabrotica species in South America. The review is a very detailed and deep revision about the topic. It contains the most important strategies used in South America to monitor and control these pests. They highlight the most abundant Diabrotica species found in South America, which have the most economical impact, and discuss about its distribution, biology, and the many different strategies used for their control. In general, while I think the text is well-written, clear and coherent, there are many instances where English could be improved as described below. I therefore suggest that an advance English speaker correct the text.

Line 43: This sentence is unclear: “Another characteristic of the North American pest Diabrotica of the fucata species group”. I would suggest: Another characteristic of the North American Diabrotica pests of the fucata species group”.

Line 48: Poaceae plants

Line 48: although the hots range…

Figure 2 and 3. Specify that the shaded area is where the insect has been reported. Also add references. Add a figure on the distribution of D. balteata.

Line 121: to hinder instead of that hinder

Line 148 and 149: unclear, please rephrase

Line 189: It is not clear if the growth regulators are used in the field and which effects they have. Please make it more clear.

In general the section 2.2 is very complete pointing out what it is known in south America. However, information on how efficient this approach is to control the pest is still missing.

Line 243: It is not clear how the intercropping information in this paragraph is related to plant resistance. Please add more information to clarify this aspect.

Line 248 : Diabrotica genus

259 and 260 : The chemical defenses of D. virgifera virgifera larvae are well studied. Please add relevant references on how benzoxazinoid sequestration protect Dvv larvae from natural enemies.

Line 269 : it seems unlikely…

Line 270 : could provide…

Line 273 : in the future…

Line 281 : third instar Diabrotica…

Line 314 : observing mating from…

Line 318 - 320 : these phrases are not clear, please rephrase.

Line 319 : change whatever to any; change state to stage

Line 321 -322 : and olfactometer experiments…

Line 324 : What is know on D. balteata sex pheromone-based traps in South America?

Author Response

First off, I would like to thank the referees on behalf of the other authors and me, for the thorough and helpful revision. We have introduced almost every modification suggested by them, which you will find in track changes throughout the manuscript.

About the advanced English speaker revision, the first and last authors of this manuscript are native English speakers (one British, one American), and fairly well published, so while we made every effort to satisfy both referees’ stylistic or grammatical demands, I confess we didn’t think it was actually necessary to get an external corrector.

Line 43: Referee suggests we write “Another characteristic of the North American Diabrotica pests of the fucata species group”.

This seems more confusing, since “Diabrotica pests” could be taken as pests of Diabrotica, or pests of the fucata group. We prefer to leave it as it is, if the editors are alright with it.

Line 48, We left “…the virgifera group feed exclusively on Poaceae”. I don’t think adding “Poaceae plants” is necessary, and it is certainly not normally seen in the literature.

The referee asks for a map of the distribution of D. balteata. Unfortunately we have no way of providing it. There are only a handful of references (two localities in Venezuela, and perhaps 4 or 5, quite close together, in Colombia) that hardly allow making a map. Nobody seems to have taken the initiative to survey this species in either country, and we haven’t found even enough references to infer their distribution. So if we drew a map with only those 7-8 points, readers may assume those are the only localities where the insect is present, and if we shaded the whole of Venezuela and Colombia, we would almost certainly be reporting areas where D. balteata is not present. We’ve added a reference to this fact in lines 75-76.

Line 148 and 149: unclear, please rephrase.

Done. Now lines 156-158.

Line 189: It is not clear if the growth regulators are used in the field and which effects they have. Please make it more clear.

This is just an inference from experimental results, it has not been applied in the field yet. We’ve clarified as much (line 196-197).

In general the section 2.2 is very complete pointing out what it is known in south America. However, information on how efficient this approach is to control the pest is still missing.

Lines 222-227 cite two publications that evaluated just that. We couldn’t come up with other study cases, or general statistics for the whole continent.

Line 243: It is not clear how the intercropping information in this paragraph is related to plant resistance. Please add more information to clarify this aspect.

It’s true, it’s not really about plant resistance, but intercropping, which may work through several different mechanisms, such as repellence or signal confusion. It can’t really go in any other section, nor deserves a section in itself, so we’ve added an introductory sentence instead (line 249), which I hope is acceptable to the referee.

259 and 260 : The chemical defenses of D. virgifera virgifera larvae are well studied. Please add relevant references on how benzoxazinoid sequestration protect Dvv larvae from natural enemies.

We’re not really writing about North American Diabrotica species, and we only mention them in passing to make a relevant point for comparison with the South American species we are reporting on. We’re reluctant to get into a discussion on DIMBOA when we simply wanted to make the rather trivial suggestion that given that North American Diabrotica larvae are known to store chemical defences, the same might be expected for South American species.

Line 319 : change state to stage

We refer to the reproductive state (i.e. virgin or mated).

Line 324 : What is know on D. balteata sex pheromone-based traps in South America?

Not one bit, I’m afraid. We didn’t find a single reference. In fact there were mighty few references to D. balteata at all in the literature, and most of them was grey literature, not peer-reviewed.

I think that’s all, the remaining corrections have been done.